# Activation of Downstream mTORC1 Target Ribosomal Protein S6 Kinase (S6K) Can Be Found in a Subgroup of Dutch Patients with Granulomatous Pulmonary Disease

**DOI:** 10.3390/cells10123545

**Published:** 2021-12-15

**Authors:** Raisa Kraaijvanger, Kees Seldenrijk, Els Beijer, Jan Damen, Jayne Louise Wilson, Thomas Weichhart, Jan C. Grutters, Marcel Veltkamp

**Affiliations:** 1Interstitial Lung Diseases Centre of Excellence, Department of Pulmonology, St. Antonius Hospital, 3435 CM Nieuwegein, The Netherlands; r.kraaijvanger@antoniusziekenhuis.nl (R.K.); e.beijer@antoniusziekenhuis.nl (E.B.); j.grutters@antoniusziekenhuis.nl (J.C.G.); 2Interstitial Lung Diseases Centre of Excellence, Pathology DNA, Department of Pathology, St. Antonius Hospital, 3435 CM Nieuwegein, The Netherlands; k.seldenrijk@antoniusziekenhuis.nl; 3Pathology DNA, Department of Pathology, Jeroen Bosch Hospital, 5223 GZ ’s-Hertogenbosch, The Netherlands; j.damen@jbz.nl; 4Center for Pathobiochemistry and Genetics, Institute of Medical Genetics, Medical University of Vienna, 1090 Vienna, Austria; jayne.wilson@meduniwien.ac.at (J.L.W.); thomas.weichhart@meduniwien.ac.at (T.W.); 5Division of Hearth and Lungs, University Medical Centre, 3584 CX Utrecht, The Netherlands

**Keywords:** sarcoidosis, mTORC1, S6K, granuloma, phenotyping

## Abstract

Mechanistic target of rapamycin complex 1 (mTORC1) has been linked to different diseases. The mTORC1 signaling pathway is suggested to play a role in the granuloma formation of sarcoidosis. Recent studies demonstrated conflicting data on mTORC1 activation in patients with sarcoidosis by measuring activation of its downstream target S6 kinase (S6K) with either 33% or 100% of patients. Therefore, the aim of our study was to reevaluate the percentage of S6K activation in sarcoidosis patients in a Dutch cohort. To investigate whether this activation is specific for sarcoid granulomas, we also included Dutch patients with other granulomatous diseases of the lung. The activation of the S6K signaling pathway was evaluated by immunohistochemical staining of its downstream effector phospho-S6 in tissue sections. Active S6K signaling was detected in 32 (43%) of the sarcoidosis patients. Twelve (31%) of the patients with another granulomatous disorder also showed activated S6K signaling, demonstrating that the mTORC1 pathway may be activated in a range for different granulomatous diseases (*p* = 0.628). Activation of S6K can only be found in a subgroup of patients with sarcoidosis, as well as in patients with other granulomatous pulmonary diseases, such as hypersensitivity pneumonitis or vasculitis. No association between different clinical phenotypes and S6K activation can be found in sarcoidosis.

## 1. Introduction

The mechanistic target of rapamycin (mTOR) is a serine–threonine protein kinase and the main component of both mTORC1 and mTORC2 complexes, which regulate different processes, such as cell growth, proliferation, autophagy, and angiogenesis [1,2]. mTORC1 also regulates activation of innate immune cells, such as monocytes, macrophages, as well as dendritic cells. Important upstream components in this mTORC1 pathway are tuberous sclerosis protein-1 & -2 (TSc1 & TSc2), which normally suppress mTORC1 activity. Two important downstream components of the mTORC1 pathway are the target protein S6 kinase (S6K), which increases the overall translation capacity of cells when activated, and eukaryotic initiation factor 4E-binding protein (4EBP1), which is crucial for ribosome recruitment [1,2,3,4]. Dysregulation of this fundamental signaling pathway has already been linked to different diseases, such as metabolic disorders, tuberous sclerosis complex, and multiple types of cancer [1]. A classic example of a pulmonary disease in which there is overactivation of mTORC1 is Lymphangioleiomyomatosis (LAM) [5]. In this disease, genetic mutations in either TSc1 or TSc2 cause an overactivation of the mTORC1 pathway, resulting in cystic destruction of lung parenchyma [6]. Furthermore, the activity of mTORC1 is frequently upregulated in human cancer. For example, in gastric cancer and non-small lung cancer, activation of mTOR pathway was found in 54% and 71% of patients, respectively [7,8], and in the latter associates with a poor prognosis [8]. 

Interestingly, activation of mTORC1 is also associated with an inflammatory-driven disease, such as sarcoidosis. Sarcoidosis is a heterogeneous, multisystemic inflammatory disorder characterized by the presence of non-caseating granuloma [9,10]. Linke et al. found that chronic mTORC1 signaling can induce macrophage granuloma formation in mice and demonstrated that activation of downstream mTORC1 target S6K could be detected in sarcoidosis patients by staining for its downstream substrate phosphor-S6 (p-S6). They found p-S6 expression in 33% of 27 sarcoidosis patients and demonstrated that this increased S6K activity was associated with disease progression [2].

Recently, Pizzini et al. studied mTORC1 activation in a cohort of 58 patients with sarcoidosis, and demonstrated that all patients had a positive activation of S6K [11], which is not in concordance with the 33% of patients described by Linke et al.

In light of the fact that not 100% of patients with other diseases, such as cancer, show activity of mTORC1, the intriguing question is why this would be different in sarcoidosis patients. Therefore, we studied S6K activation in a cohort of Dutch patients with sarcoidosis. We hypothesize that the mTORC1 signaling pathway is not active in all patients with sarcoidosis. Furthermore, as mTORC1 is not disease-specific, we hypothesize that this signaling pathway will also be activated in other granulomatous diseases of the lung in a subgroup of patients.

## 2. Materials and Methods

### 2.1. Study Population

Unstained tissue blocks of two study cohorts previously studied at the St. Antonius Hospital (Nieuwegein, The Netherlands) were requested; one cohort contained 76 sarcoidosis patients [12] and the other cohort consisted of 48 patients with other granulomatous disorders [13]. The diagnosis of sarcoidosis had been established according to the criteria of the American Thoracic Society/European Respiratory society [14]. Patients were included in the study when enough residual tissue was available and when presence of granulomas could be detected in the hematoxylin and eosin (HE)-stained tissue sections. As a positive control, we included 6 patients with LAM based on the fact that an increased mTORC1 activity must be present in this condition.

The study was approved by the Medical research Ethics Committees United (MEC-U) of the St. Antonius Hospital (R05-08A) and written consent was obtained from all patients.

### 2.2. Immunohistochemistry

Formalin-fixed paraffin-embedded tissue sections (4 µm thick) were immunohistochemically stained with the phosphor-S6 ribosomal protein (p-S6) (Ser240/244) (DD68F8) XP^®^ Rabbit mAB (Cell Signaling Technology, Inc., Danvers, MA, USA). We followed to protocol described by the manufacturer; however, instead of the manual procedures, the sections were stained with the use of a VENTANA BenchMark ULTRA (Ventana Medical Systems, Inc., Tucson, AZ, USA). We modified the original protocol to optimize the sensitivity and specificity of the staining results for the VENTANA BenchMark ULTRA. Shortly, sections were de-paraffinized and rehydrated followed by antigen retrieval using ULTRA CC2, (Ventana Medical Systems, Inc., Tucson, AZ, USA) for 24 min at 95 °C. The p-S6 antibody was used in a concentration of 1:800 (diluted with Ventana antibody diluent, REF 251-018, Ventana Medical Systems, Inc., Tucson, AZ, USA) and incubated for 32 min at room temperature. Furthermore, a detection kit with ultraView was used (REF 760-500, Ventana Medical Systems, Inc., Tucson, AZ, USA). Sections were counterstained with Mayer’s hematoxylin (REF 760-2021, Ventana Medical Systems, Inc., Tucson, AZ, USA) for 4 min at RT, followed by Bluing Reagent (REF 760-2037, Ventana Medical Systems, Inc., Tucson, AZ, USA) for 4 min at RT. All operations were performed on the machine (on board). Dehydration was performed by successive alcohol 96%, 2 times alcohol 100%, and 2 times xylene. Tissue slides were covered with tape using cover-slipper Tissue-Tek (SAKURA Fineteck USA Inc Torance, CA, USA).

### 2.3. Evaluation of Immunohistochemistry

The results of the immunohistochemical staining for p-S6 were analyzed by a pulmonary pathologist with more than 25 years of experience in interstitial lung diseases (KS). The grading system was based on the percentage of p-S6 positive cells within granulomas and their intensity, as previously described [15]. Staining intensity of p-S6 was scored as: 0 (negative), 1 (weak), 2 (moderate), or 3 (strong). An example of the staining intensities is shown in Figure 2. The percentage of p-S6 positive cells was scored as: 0 (<10%), 1 (10–25%), 2 (25–50%), 3 (50–75%), or 4 (>75%). The histological score was generated as the sum of intensity and percentage of positive cells, ranging from 0 to 7.

### 2.4. Identification of p-S6 Related Phenotypes

To determine possible clinical phenotypes related to the percentage of S6K activation, basic characteristics (age at diagnosis, gender, co-morbidities), the Scadding stage at diagnosis and follow up, the presence of Löfgren syndrome, as well as information on therapy and organ manifestation were collected from medical records of the patients. The disease status of patients was retrospectively examined and classified according to the Clinical Outcome Status (COS) [16] 2 and 5 years after diagnosis, COS categorizes patients into 9 scores: (1) resolved disease, never treated; (2) resolved disease, no therapy > one year; (3) minimal disease, never treated; (4) minimal disease, no therapy > one year; (5) persistent disease, never treated; (6) persistent disease, no therapy > one year; (7) persistent disease, current therapy but no worsening in prior year and asymptomatic; (8) persistent disease, current therapy but no worsening in prior year and symptomatic; and (9) persistent disease, current therapy which worsened in the prior year. Sarcoidosis patients were classified into two phenotype groups: A—resolved, minimal, or persistent disease without treatment (COS 1-6), and B—persistent disease with need for treatment (COS 7-9) [12].

### 2.5. Statistical Analysis

Data were analyzed using IBM SPSS statistics version 24. An unpaired T-test was used to compare numerical data. Non-parametric tests were used for non-normally distributed data (Mann–Whitney U test). Categorical variables were compared using Chi-Squared and Fisher’s exact test, where appropriate. *p*-values < 0.05 were considered significant.

## 3. Results

### 3.1. Active mTORC1 Signaling Present in All Six Patients with Lymphangioleiomyomatosis

In total, six patients with LAM were included in the immunohistochemical staining. All patients were female, with a mean age of 44.56 ± 15.89 years. At the time of biopsy none of the patients received medication. Eventually, four (66.7%) of the patients needed treatment with the mTORC1 inhibitor Sirolimus. In two (33.3%) patients, a mutation in TSc2 was established. LAM was diagnosed according to immunohistochemical staining with human melanoma black (HMB45). At the regions positive for HMB45, the p-S6 antibody also showed to have a positive expression, as shown in Figure 1. Tissue samples of all (100%) of the patients showed a positive p-S6 expression at the regions of HMB45 expression. Intensity of p-S6 was highest for the two patients with a mutation in TSc2.

### 3.2. Characteristics of Sarcoidosis Patients

Formalin-fixed paraffin-embedded tissue blocks were available from 74 patients with sarcoidosis. Detailed information about the sarcoidosis cohort is shown in Appendix A. The mean age of the sarcoidosis patients was 44.90 ± 12.47 years and 55% were male. In total, among 50 patients (67.6%) who presented with extra pulmonary manifestation, the most common involvements were lymph nodes (*n* = 23, 31%), skin (*n* = 18, 24%), nerve system (*n* = 13, 18%), and heart (*n* = 12, 16%). Löfgren syndrome was present in two (3%) patients and, radiologically, Scadding stage 2 was the most frequent at the time of biopsy.

At time of biopsy, seven patients received treatment (four patients received prednisone mono-therapy, one patient received prednisone as well as methotrexate, one patient received prednisone and azathioprine, and one patient received azathioprine mono-therapy). Respectively, 65% and 43% were classified in phenotype group B, 2 and 5 years after diagnosis.

### 3.3. Not All Patients with Sarcoidosis Demonstrate Active mTOR Signaling

Figure 2 shows the different intensities of the p-S6 antibody that could be found in the immunohistochemical staining. In 32 (43.2%) of the sarcoidosis patients, p-S6 expression was found within the granulomas. The mean IHC-score for the sarcoidosis patients was 0.92 ± 1.31 (Table 1). No correlation could be found between p-S6 expression and ACE and sIL-2R levels.

When comparing patients with and without medication at time of biopsy, no significant differences in mTORC1 activity was observed (*p* = 0.435) (Table 2). Similar findings could be described when comparing patients with and without extra pulmonary manifestation (*p* = 0.630), as well as when comparing the four Scadding stages at time of biopsy (*p* = 0.119). Furthermore, no specific phenotype could be identified when looking at persistent disease, according to the COS-score, 2 and 5 years after diagnosis (*p* = 0.346 and *p* = 0.216), and there was no need for third-line therapy (*p* = 0.801).

### 3.4. mTORC1 Activity Is Also Present in a Subgroup of Patients with Hypersensitivity Pneumonitis and Vasculitis

The control group of other granulomatous disorders consisted of 29 patients with hypersensitivity pneumonitis (HP), 7 patients with granulomatosis with polyangiitis (GPA), 1 patient with eosinophilic granulomatosis with polyangiitis (EGPA), 1 patient with tuberculosis, and 1 patient with pneumoconiosis. This cohort is slightly older than the sarcoidosis cohort (52.40 ± 11.47), and the amount of Caucasians is higher (97.4%). The amount of males and history of smoking is similar. At time of biopsy, five patients received medication (four patients received methotrexate and one patient received Cellcept). The characteristics of this group are stated in Appendix A.

In 12 (30.8%) of the patients in this group, p-S6 expression was found within the granulomas with a mean IHC-score of 0.67 ± 1.13 (Table 1). Figure 3 shows immunohistochemistry staining in tissue of a patient with GPA and of a patient with HP. Comparing the p-S6 positivity, no significant difference (*p* = 0.628) was observed between the sarcoidosis group and the other granulomatous disorders.

## 4. Discussion

Previous studies already suggested a possible role for the cellular signaling pathway mTORC1 in the granuloma formation of sarcoidosis. However, the results from these studies conflicted each other, compelling us to address the question in which percentage of sarcoidosis patients mTORC1 might play a role in granuloma formation and/or maintenance.

Our data reveal that in approximately 43% of Dutch patients with sarcoidosis, activation of the mTORC1 pathway measured by the downstream kinase S6K can be detected inside granulomas. These results are consistent with the data from Linke et al., where activation of S6K could also be demonstrated in only a subgroup of patients (33%).

However, in the recent paper from Pizzini et al., it was demonstrated that S6K was activated in 100% of the 58 patients with sarcoidosis. An explanation for the discrepancy was lacking, although Pizinni et al. suggested differences in clinical presentation, cohort size, and diverging definitions of IHC stains as possible confounders. Our cohort consisted of 74 patients with sarcoidosis, including different clinical presentations, and IHC staining was scored using the method described by Pizinni et al. Therefore, in our opinion, the aforementioned possible confounders are unlikely to explain the difference in sarcoidosis patients with S6K activation.

A possible explanation for the differences could be the different antibodies used. For the immunohistochemical staining of S6K, anti-p-S6 (ser240/244) was used in our study as well as in the study of Link et al. Pizzini et al., which used anti-p-S6 (ser235/236), which stains for a different phosphorylation site of the S6 protein. Previous studies have shown that the ser240/244 residue can only be phosphorylated via the mTORC1/S6K signaling pathway, while the ser235/236 residue can also be phosphorylated independently of the mTORC1/S6K signaling pathway [4,17]. Pizzini et al. also stained the other downstream substrate of mTORC1, 4E-BP1, in order to more accurately address mTORC1 activation. However, Linke et al. demonstrated that, after treating cells derived from TSC2 knock-out mice with rapamycin, 4E-BP1 was still activated while p-S6 was not, suggesting that 4E-BP1 can also be activated independently of the mTORC1 signaling pathway. A comparison between our data and that of Pizzini et al. could suggest that, in our study, activation of S6K may be underestimated in patients with sarcoidosis, while in the Pizzini paper, activation of S6K could possibly be overestimated. The fact that all of our six patients with LAM stained positive for S6K in our study suggests that the technical aspects of our S6K staining are reliable.

Regarding the absence of an association between S6K activation and different clinical phenotypes, our data are in concordance with the data of Pizzini and colleagues. It is important to state that S6K activity was assessed in biopsies taken at the time of diagnosis and not during the disease course. At present, it is unknown whether the activity of the mTORC1 signaling pathway is dynamic during the course of sarcoidosis or whether it serves as an on–off switch at the beginning of the disease when granuloma formation is initiated. Further insight in the mechanistic role of mTORC1 seems important if we want to implement mTORC1 inhibition in the treatment of patients with sarcoidosis. In oncology, it is generally accepted that during the course of the disease, new biopsy material is sometimes obtained to re-assess (genetic) characteristics of the tumor. A next step in sarcoidosis research, therefore, could be to determine mTORC1 activity in patients at multiple times during the course of their disease. This may provide additional information about a patient’s likelihood of responding when initiated on mTOR inhibitors as a last-resort therapy.

Based on the suggested role of mTORC1 signaling pathway as a molecular mechanism in the initiation and maintenance of granulomas [2], it is not surprising to find S6K activation in a subgroup of patients with other pulmonary granulomatous diseases, such as HP and vasculitis. The fact that mTORC1 activation is also found in a subgroup of patients with malignancies, such as gastric cancer or lung cancer, together with our observation that 32% of the patients with HP and vasculitis show S6K activation, supports the hypothesis that mTORC1 is probably not universally active in different diseases. To our knowledge, this is the first study where S6K activation in granulomatous diseases, other than sarcoidosis, is investigated.

Our findings can be particularly interesting for patients with fibrosing HP due to the fact that diagnosis is often obtained later in the disease course when pulmonary fibrosis is already present. With the new concept of progressive fibrosing ILD (PF-ILD) in mind, it may be useful to look at mTORC1 activation in HP patients in order to gain arguments for increasing immunosuppressive therapy using mTORC1 inhibitors in addition to anti-fibrotic therapy.

Our study has some limitations. First, the study is of retrospective design. As most tissue sections were derived from other hospitals in the Netherlands, not all characteristic information and follow-up data could be collected. Consequently, only limited data were available on biomarker levels and the COS scores after five years could not be determined in all patients. Moreover, the majority of the sarcoidosis patients in our cohort probably have a more severe form of sarcoidosis compared to other hospitals, since the Antonius Hospital is a national referral center for ILD and Sarcoidosis. As a consequence, we had very few patients in the resolved and minimal disease COS groups, making it impossible to adequately analyze whether disease status alone, irrespective of use of medication, was associated with mTORC1 activation.

Another limitation is that we were not able to analyze the disease course in the patient group with other granulomatous disorder due to the small sample size in combination with too much missing follow-up data. Consequently, we were unable to examine whether mTORC1 activation is associated with progressive disease in this group.

## 5. Conclusions

Activation of downstream mTORC1 target ribosomal protein S6 kinase (S6K) can be found in a subgroup of patients with sarcoidosis, as well as in patients with other granulomatous pulmonary diseases, such as hypersensitivity pneumonitis or vasculitis. Even though no association between different clinical phenotypes and S6K activation could be found in sarcoidosis, the subdivision of sarcoidosis patients by S6K activation detected in the granuloma could possibly benefit the personalized medicine of this disease regarding choice of immunosuppressive therapy.

## Figures and Tables

**Figure 1 cells-10-03545-f001:**
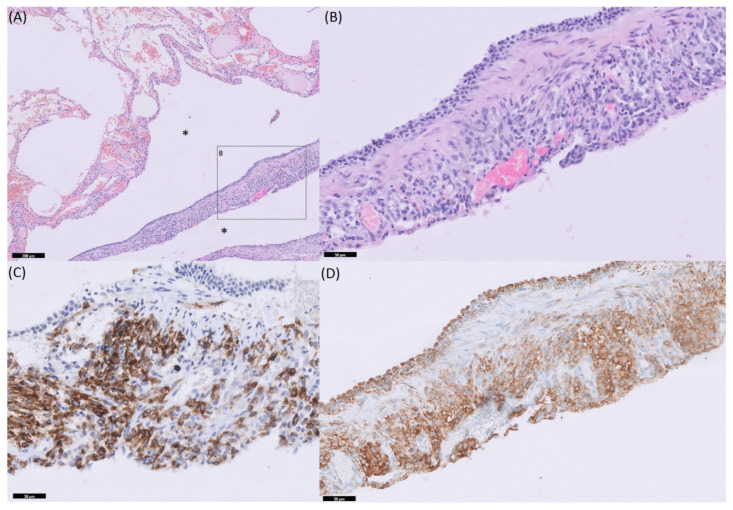
Example of Immunohistochemistry (IHC) staining in lung tissue of a patient with lymphangioleiomyomatosis (LAM): (**A**) Hematoxylin-eosin (HE) staining. Upper part of (**A**) represent normal lung parenchyma, lower part demonstrates affected tissue. Asterisks represent cystic space. (**B**) Higher magnification of area B from (**A**), (**C**) human melanoma black (HMB45) staining, and (**D**) p-S6 (S6K activity) staining.

**Figure 2 cells-10-03545-f002:**
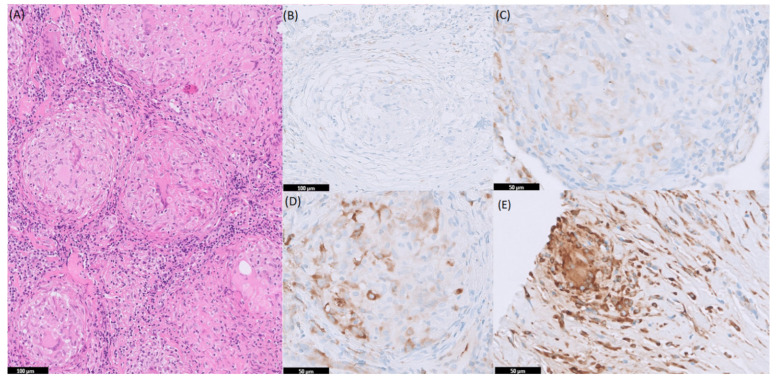
p-S6 (S6K activity) immunohistochemistry (IHC) staining in sarcoidosis tissue (**A**) Hematoxylin-eosin (HE) staining of tissue from lymph node, (**B**) negative staining of tissue from lung, (**C**) weak staining of tissue from lung, (**D**) moderate staining of tissue from lymph node, and (**E**) strong staining of tissue from skin.

**Figure 3 cells-10-03545-f003:**
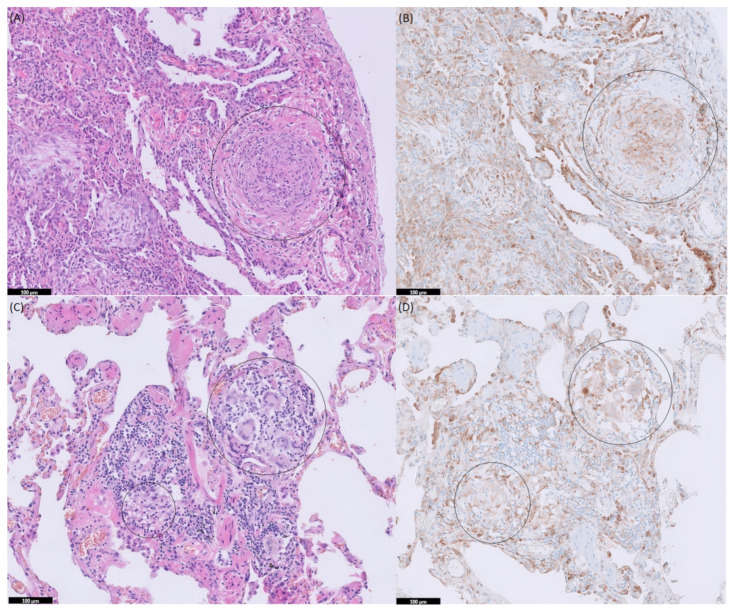
Immunohistochemistry (IHC) staining in lung tissue of patient with granulomatosis with polyangiitis (GPA); (**A**) Hematoxylin-eosin (HE) staining and (**B**) p-S6 (mTORC1 activity) staining, and in lung tissue of patient with hypersensitivity pneumonitis (HP); (**C**) HE staining and (**D**) p-S6 (S6K activity) staining. Circled area represents granuloma.

**Table 1 cells-10-03545-t001:** Immunohistochemical (IHC) scoring of percentage of p-S6 positive cells within the granuloma of sarcoidosis patients and of patients with other granulomatous disorders, ranging from 0 to 7. Data are shown as absolute numbers with percentage in brackets.

IHC-Score	Sarcoidosis (*n* = 74)	Other Gran. Disorders (*n* = 39)	*p*-Value
0	42 (56.8)	27 (69.2)	0.198
1	13 (17.6)	3 (7.7)	0.154
2	8 (10.8)	5 (12.8)	0.751
3	6 (8.1)	3 (7.7)	0.938
4	4 (5.4)	1 (2.6)	0.487
5	1 (1.4)	0 (0)	0.468
6	0 (0)	0 (0)	-
7	0 (0)	0 (0)	-

**Table 2 cells-10-03545-t002:** Comparison of p-S6 scores in sarcoidosis patients according to medication use, need for third-line therapy, inflammatory biomarkers and organ involvement. The negative group consists of patients with IHC-score of 0, the positive group consists of patients with IHC-score above 0.

Parameter	Negative (*n* = 42)	Positive (*n* = 32)	*p*-Value
Löfgren syndrome	2	0	0.231
Medication at time of biopsy	3	4	
Prednisone	2	4	0.235
Methotrexate	0	1	0.550
Azathioprine	1	1	0.630
Third-line therapy	14	12	0.801
COS group 2 years follow up (1/2/3)	(3/11/26)	(0/8/22)	0.346
COS group 5 years follow up (1/2/3)	(2/9/17)	(0/2/15)	0.216
Scadding stage at time biopsy (0/I/II/III/IV)	(1/11/15/6/7)	(2/10/16/3/1)	0.119
Inflammatory biomarkers ^a^			
ACE (U/L)	5	5	0.498
sIL-2R (pg/mL)	5	8	0.102
Number of other organs involved ^b^ (0/1/2/3/4)	(15/20/2/3/2)	(15/9/3/5/0)	0.691
Skin	7	5	0.905
Eyes	3	3	0.869
Liver	5	3	0.809
Heart	10	2	0.052
Spleen	1	2	0.152
Bones	2	2	0.821
Neurosarcoidosis	8	5	0.752
Small Fiber Neuropathy	8	1	0.069

^a^ inflammatory biomarkers consist of patients with increased concentrations of ACE and sIL-2R. ^b^ amount of involved organs different from the organ from which a tissue biopsy was taken.

## Data Availability

The data presented in this study are available on request from the corresponding author.

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
