# Peer review of "Activation of Downstream mTORC1 Target Ribosomal Protein S6 Kinase (S6K) Can Be Found in a Subgroup of Dutch Patients with Granulomatous Pulmonary Disease"

_cells, 2021, doi:10.3390/cells10123545_

Round 1

Reviewer 1 Report

In this paper from Kraaijvanger R et al, the authors indicate that in a Dutch cohort suffering of sarcoidosis and evaluate the percentage of S6K activation. Furthermore in this study, the authors include other granulomatous diseases of the lung. A major concern have arisen after reading the manuscript in its present form:

1. Although the experiments are correctly performed, there is a very important control to include in the present study. Throughout the manuscript, the authors analyze phospho-S6 as a marker of mTORC1 activation and this is not completely correct. It is true that under a mitogenic event, S6 is phosphorylated by multiple kinases, including p70S6K (which derives from mTORC1). For that reason, the authors must analyze, in parallel to the phospho-S6 antibody, the phosphorylation status of p70S6K at Threonine 389, which is specific residue for measuring mTORC1 activity directly. 

Author Response

We thank the reviewer for taking the time to read our manuscript. We agree with the reviewer that p70S6K (Thr389) is a specific residue for mTORC1 activity and would be an appropriate marker for mTORC1 activity.  However, we chose to stain our biopsy lesions for anti-pS6(Ser240/244) for several reasons. First of all, we wanted to compare our results with the initial paper of the group by Linke et al published in Nature Immunology.  Secondly, we chose to stain with only one monoclonal antibody based on the fact that 4E-BP1 can be activated independently of the mTORC1 signaling pathway. In the paper of Linke et al, western blot analysis revealed that when treating cells of TSC2-knock-out mice with rapamycin, p-S6 was no longer detectable, in contrast to 4E-BP1 which could still be detected. More importantly regarding the suggestion of the reviewer, is the fact that another study showed that when treating cells with rapamycin phosphorylation of 70S6K (Thr389) was suppressed  resulting in no phosphorylation at site S6 (ser240/244) [1].

Adding the suggested staining to our manuscript would, in our opinion, not change one of the messages of our manuscript that in our study activation of S6K could be underestimated in patients with sarcoidosis while in the Pizzini paper activation of S6K could possibly be overestimated

[1]         M. W. Edwards et al., “Role of mTOR downstream effector signaling molecules in Francisella tularensis internalization by murine macrophages,” PLoS One, vol. 8, no. 12, pp. 1–20, 2013.

Reviewer 2 Report

The authors have satisfactorily addressed previously raised concerns except the one to exclude the possibility that increased pS6 staining is due to increased total S6 expression instead of increased S6K activity. This would need to be either experimentally addressed by providing total S6 staining, or discussed in the manuscript.

Author Response

We thank the reviewer for taking the time to read our manuscript. We agree that S6 expression may be increased independently of S6K activity. As previous research showed, the S6 protein can be activated at different phosphorylation sites, as a result of different activation routes. However different studies have shown that the antibody that is used in our study S6 (Ser240/244) is not phosphorylated independently of S6K activity [1], [2]. For a more complete view on the role of the mTORC1 in sarcoidosis the two downstream targets should be stained in one biopsy lesion, as was done by Pizzini et al. We address this topic in our discussion together with the hypothesis that our staining may perhaps be an understatement of the total S6K activity while in the Pizzini paper activation of S6K could possibly be overestimated. We therefore decided not to include additional staining in our manuscript.

[1]         M. Pende et al., “ S6K1 −/− / S6K2 −/− Mice Exhibit Perinatal Lethality and Rapamycin-Sensitive 5′-Terminal Oligopyrimidine mRNA Translation and Reveal a Mitogen-Activated Protein Kinase-Dependent S6 Kinase Pathway ,” Mol. Cell. Biol., vol. 24, no. 8, pp. 3112–3124, 2004.

[2]          P. P. Roux et al., “RAS/ERK Signaling Promotes Site-specific Ribosomal Protein S6 Phosphorylation via RSK and Stimulates Cap-dependent Translation,” vol. 18, no. 11, pp. 1492–1501, 2011

Round 2

Reviewer 1 Report

The authors have answered all the questions arisen by this referee